# The Evidence-Based Development of an Intervention to Improve Clinical Health Literacy Practice

**DOI:** 10.3390/ijerph17051513

**Published:** 2020-02-26

**Authors:** Gill Rowlands, Bimasal Tabassum, Paul Campbell, Sandy Harvey, Anu Vaittinen, Lynne Stobbart, Richard Thomson, Mandy Wardle-McLeish, Joanne Protheroe

**Affiliations:** 1Population Health Sciences Institute, Baddiley Clark Buildig, Newcastle University, Newcastle upon Tyne NE2 4AX, UK; bimasal@icloud.com (B.T.); a.m.vaittenen@outlook.com (A.V.); lynnestobbart@sky.com (L.S.); richard.thomson@newcastle.ac.uk (R.T.); 2Faculty of Medicine and Health Sciences, Keele University, Staffordshire ST5 5BG, UK; p.campbell@keele.ac.uk (P.C.); j.protheroe@keele.ac.uk (J.P.); 3Patient Research Ambassador, (North East and North Cumbria) and Voice Research Advisor, Voice, Newcastle upon Tyne NE1 4BF, UK; 4Community Health and Learning Foundation, currently Reaching People, 15 Wellington Street, Leicester LE1 6HH, UK; mandy.wardle@ntlworld.com

**Keywords:** health literacy, intervention, primary care, feasibility study

## Abstract

Low health literacy is an issue with high prevalence in the UK and internationally. It has a social gradient with higher prevalence in lower social groups and is linked with higher rates of long-term health conditions, lower self-rated health, and greater difficulty self-managing long-term health conditions. Improved medical services and practitioner awareness of a patient’s health literacy can help to address these issues. An intervention was developed to improve General Practitioner and Practice Nurse health literacy skills and practice. A feasibility study was undertaken to examine and improve the elements of the intervention. The intervention had two parts: educating primary care doctors and nurses about identifying and enhancing health literacy (patient capacity to get hold of, understand and apply information for health) to improve their health literacy practice, and implementation of on-screen ‘pop-up’ notifications that alerted General Practitioners (GPs) and nurses when seeing a patient at risk of low health literacy. Rapid reviews of the literature were undertaken to optimise the intervention. Four General Practices were recruited, and the intervention was then applied to doctors and nurses through training followed by alerts via the practice clinical IT system. After the intervention, focus groups were held with participating practitioners and a patient and carer group to further develop the intervention. The rapid literature reviews identified (i) key elements for effectiveness of doctors and nurse training including multi-component training, role-play, learner reflection, and identification of barriers to changing practice and (ii) key elements for effectiveness of alerts on clinical computer systems including ‘stand-alone’ notification, automatically generated and prominent display of advice, linkage with practitioner education, and use of notifications within a targeted environment. The findings from the post-hoc focus groups indicated that practitioner awareness and skills had improved as a result of the training and that the clinical alerts reminded them to incorporate this into their clinical practice. Suggested improvements to the training included more information on health literacy and how the clinical alerts were generated, and more practical role playing including initiating discussions on health literacy with patients. It was suggested that the wording of the clinical alert be improved to emphasise its purpose in improving practitioner skills. The feasibility study improved the intervention, increasing its potential usefulness and acceptability in clinical practice. Future studies will explore the impact on clinical care through a pilot and a randomised controlled trial.

## 1. Introduction

Health literacy can be defined as ‘the motivation, knowledge and competencies to access, understand, appraise and apply health information in order to make judgments and take decisions in everyday life concerning health care, disease prevention and health promotion to maintain or improve quality of life throughout the course of life’ [1]. Health literacy can be conceptualised as involving a range of capacities including functional (reading and writing skills as applied to health), interactive (skills to actively participate in everyday activities, extract information and derive meaning from different forms of communication, and to apply new information to changing circumstances), and more advanced skills to critically analyse information, and to use this information to exert greater control over life events and situations [2]. The importance of health literacy is increasingly being recognised due to the prevalence of low health literacy and to the associations between low health literacy and poorer health, greater illness, and increased health care costs [3,4]. In England, the data available on population health literacy measure only functional skills. The prevalence of low functional health literacy in the English working-age population is 43%, rising to 61% when understanding health information involves numeracy skills [5]. Low health literacy is associated with lower self-rated health and higher rates of long-term health conditions [3,4]. Furthermore, people with inadequate and problematic health literacy and a chronic health condition find their health condition more limiting than people with adequate health literacy and a chronic health condition, and people with lower health literacy have a greater frequency of contacts with the health service than people with higher health literacy [4].

Health literacy is a balance between peoples’ capacities and the complexities of the health system [6]; thus, improving health literacy awareness and skills in health care organisations and those working in them could contribute to an increase in patients’ capacities for health [7]. Furthermore, such ambitions fit with UK Government policy to increase patient participation in health care and increase informed choice of health care options [8,9] General practices and other primary and community care settings are ideally placed in this regard; in the UK, approximately 95% of health service contacts take place in general practices [10].

In England, nearly all general practices use electronic health records (EHRs) for clinical care [11]. Increasingly, clinicians and health service managers are using automatically generated clinical alerts [12]; such alerts have been shown to improve care, either in complex systems providing clinical support across multiple settings [13] or in relation to specific health conditions such as antibiotic prescribing for upper respiratory tract Infections [14]. 

Previous work, using an English working-age population functional health literacy dataset, has determined a threshold of literacy and numeracy skills required to understand and use health information in common circulation in the UK [5]. Building on this, an algorithm has been developed to calculate the probability that any individual falls below this threshold. The algorithm uses data available in UK general practice i.e., age, ethnicity, first language and small geographical area of residence (Lower Super Output Area: LSOA). This algorithm can be run automatically in EHR systems to generate a clinical (‘pop-up’) alert when the notes of a patient with a high probability of falling below the competency threshold are accessed [15]. Separately, general health literacy training for GPs has also been developed and delivered [16]. 

The current study aimed to build on this previous research by (i) developing an evidence-based intervention that incorporates health literacy awareness training to General Practitioners and Practice Nurses and applies an automatically generated health literacy alert within the General Practice EHR system, (ii) investigating the feasibility of this intervention from the perspectives of participating health care staff and a patient and carer group, and (iii) gathering the views of practitioners and members of a patient and carer group on how the intervention could be improved. The aim of the intervention would be to improve practitioner health literacy awareness, skills, and practice to improve the patient experience.

## 2. Materials and Methods

### 2.1. Ensuring the Intervention Was Evidence Based

In order to ensure that the intervention components reflected best current knowledge, two rapid reviews were undertaken, one for each component. Rapid reviews aim to retain the rigorous approach undertaken in systematic reviews whilst streamlining the approach to enable identification and synthesis of evidence to be undertaken in a timely manner [17]. The rapid reviews followed the approach outlined Khangura et al. [17]. In both reviews, a key document was identified by experts to help guide the search strategy and to assess its effectiveness. These consisted of a systematic review on the effects of continuing clinical education meetings and workshops on professional practice and outcomes [18] and a systematic review on the effects of EHR clinical alerts on processes and outcomes of care [19]. 

The rapid reviews were limited to peer-reviewed publications available in English. Papers that described a protocol only were excluded. Since both key papers were published in 2009, the search was undertaken from 2008 onwards to the date of the searches (2017). Both searches were designed and tested in MEDLINE (the National Library of Medicine (US) database of journal citations and abstracts for biomedical literature) and then translated and run in the following databases: MEDLINE, Embase (the Excerpta Medica Database), CINAHL (the Cumulative Index to Nursing and Allied Health Literature) and EPOC (the Effective Practice and Organisation of Care). Example MEDLINE search strategies for the two rapid reviews are shown in Table A1 and Table A2. Because of the high number of hits produced by the two searches, a decision was made to restrict inclusion to the ‘highest’ level of evidence, i.e., systematic reviews [20]. All abstracts were read by two members of the research team, with discussion and agreement where there was a difference of opinion. Full papers were read by two members of the research team and discussed within meetings with experts. Separate meetings were held with experts in GP professional training and in electronic health record systems. Data extraction tables were developed that included key outcomes, facilitators of effectiveness, barriers to effectiveness, occasions where no effect was seen, and comments. The data extraction tables were used to identify key aspects of the included papers, which were incorporated into the intervention. 

### 2.2. Intervention

At the commencement of the intervention development, it was decided to have two components (1) GP and practice nurse training in health literacy and (2) automatically generated ‘pop-up’ alerts appearing on the GP clinical system consultation screen. The GP and practice nurse training was based on one-day health literacy training sessions that had already been shown to improve practitioner knowledge and confidence [16]. To make the training more attractive to busy practitioners, the length of the training was reduced to 3 hours. The training was face to face and involved a mixture of didactic teaching and practical exercises. 

### 2.3. Undertaking Feasibility Testing of the Intervention and Study Components

#### 2.3.1. Practice Recruitment

The IT system coding to generate the clinical alerts was written for SystmOne, a clinical IT system used by more than 2700 of the 7000 General Practices in England [21]. All practices in Newcastle and Gateshead Clinical Commissioning Group (CCG )using this system were approached to participate. Recruitment was via a letter from the Newcastle GP representative on the project steering group, and via the CCG practice newsletter. The intervention was then run in the participating practices for two months. 

#### 2.3.2. Post-Hoc Improvement of the Intervention

GP evaluation: Post-intervention focus groups were conducted in two of the four participating practices to explore key elements of the intervention and study design. Questions explored the acceptability of the study to GPs; ways to encourage participation; the feasibility of the intervention; the GP training and how it might be improved; whether patients or GPs might find elements of the intervention unhelpful and/or intrusive and, if so, how this might be addressed; and the optimal threshold for ‘pop-up’ frequency.

Service User Evaluation. The results of the study were presented to members of Voice, a Newcastle-based service user and carer organisation [22]. Views were sought on the acceptability of the process, the impact of a patient potentially being able to view a pop-up during the consultation, and whether these are stigmatizing or in any other way unpleasant, and what might be done to address this. 

### 2.4. Ethical Issues

NHS Ethics approval was obtained from the London City & East Research Ethics Committee reference 17/LO/1277. Low literacy and numeracy carry considerable stigma [23]. We sought to minimise this by emphasising to GPs that the pop-ups did not ‘diagnose’ low health literacy, but instead were intended as an alert for them to consider health literacy issues in the consultation. 

### 2.5. Analysis

This was descriptive and included:The outcome of the rapid evidence scans of (a) health professional training and (b) factors facilitating the impact of clinical system alerts on clinical practice;Practice recruitment;Qualitative thematic analysis of the post-hoc focus groups of (a) primary care practitioners and (b) patients and carers.

## 3. Results

### 3.1. Rapid Evidence Scans

#### 3.1.1. Approaches to Health Professional Training Shown to Improve Clinical Practice

The flow chart for this rapid review is shown in Figure A1. The search strategy identified 1311 articles, which were reduced to 1255 articles after duplicates were removed. After abstract review, 24 full papers were considered for inclusion, of which nine were included. 

Key facilitators for effective health professional training relevant to the intervention being developed were inclusion of mixed teaching methods such as a mixture of didactic teaching and practical/participatory methods [18,24,25,26,27,28,29,30]; including use of simulated patients [24] and small group learning [24]; building on practitioners current skills and knowledge [27]; practitioner commitment to change practice [29,30] and identifying and discussing ways of addressing barriers to change [27]; and involving local opinion leaders [28]. One paper reported systematic identification of patients most likely to benefit [31] as being a facilitator for change.

Mono-component training was found to be less effective than multi-component teaching, particularly when didactic-only [26], and this was also the case for training where either the condition or the intervention was complex [18,25], or where conditions were considered not to be serious, by those attending the courses [18]. Poor attendance and engagement with the training also resulted in lower effectiveness [25].

#### 3.1.2. Factors Facilitating the Impact of Clinical System Alerts on Clinical Practice

The flow chart for this rapid review is shown in Figure A2. The search identified 1886 articles, which were reduced to 1735 articles after duplicates were removed. After abstract review, 35 full papers were considered for inclusion, out of which four were included.

Elements of clinical alerts that promoted effectiveness were those that were directed at changing clinician behavior [32]; available during the clinical encounter [32]; specific to the situation [12]; and specific to the patient rather than the condition [32]. Clinical alerts were more effective where they required a positive action such as noting agreement [12,19,33]. Stand-alone alerts were more effective than those involving complex multi-stage interventions [19,33]. Finally, automatic (‘push’) alerts were more effective than alerts generated by the user (‘pull’ alerts) [19]. High frequency of alerts reduced effectiveness [33].

### 3.2. Integration of Rapid Review Findings into the Intervention

The training element of the intervention [16] already used mixed-methods teaching including lectures and small group work, built on current knowledge, and worked with practitioners to identify and overcome barriers to changing practice. By its nature, the 2-component intervention involved systematic identification of patients most likely to benefit (through the clinical alerts). In the light of the rapid review findings, a local opinion-leader was involved in the training, and the small group work was widened to include role play simulating consultations with patients with low health literacy, where the practitioners took it in turns to be the ‘practitioner’ and the ‘patient’, and utilised practical techniques to improve health literacy practice, such as Teach-Back [34] and ‘chunk and check’ [35].

The clinical alert component of the intervention was being developed de novo and was thus designed to include all the elements promoting effectiveness. The alerts were designed to remind the clinicians to improve their health literacy practice, as learned in the training session. The alerts automatically appeared during the clinical encounter. To maximise specificity of the alerts, the algorithm was triggered if the patient characteristics indicated a 70% or greater likelihood that the patient would fall below the national health literacy thresholds, developed in earlier work by GR et al. [5]. The alert required the clinicians to note that they had seen it, at which point it disappeared from the screen. The alerts were stand-alone. The prevalence of low functional health literacy in the general population in the UK is 61% [5], and, given the association of low health literacy with lower health and greater illness [3,4], the prevalence amongst patients attending for health consultations is expected to be even higher than this. To reduce the likelihood of ‘alert fatigue’ [33], alerts only activated during consultations for review of long-term health conditions, where health literacy is known to play a crucial role as there is greater focus on patient self-management, and where there is additional time reserved for the consultation which would allow the clinicians more time to adapt their communication style. The alerts were active over an eight-week period. The text in the alerts was ‘consider health literacy in this consultation’.

### 3.3. Practice Recruitment and Delivery of the Intervention

Twelve general practices were eligible as they used the SystmOne clinical system. All were approached. Four practices were recruited, giving a recruitment rate of 33%. Training for three practices was delivered in one group teaching session, and one practice received training via a webinar. Following the training (within two weeks) the clinical alerts were activated in the practices for eight weeks.

#### 3.3.1. Post-Hoc Improvement of the Intervention

##### Feedback from the General Practice Staff

Focus groups were undertaken at two of the four participating practices. In both, feedback was that the intervention had worked smoothly, the training was well received and effective, and that clinical alerts appeared as expected in a corner of the screen when patient notes were accessed. 

##### Issues Identified by Practitioners

There was positive feedback on the clinical alerts; participants reported that the alerts reminded them about the training they had received: “it definitely jogged our memory when we saw it”, “it did make me for a few seconds think “Right, I just need to try and speak to them appropriately””. Practitioners did, however, raise some hypothetical concerns, i.e., that patients could be offended if they became aware of the alerts during the consultation. The length of time for which the clinical alerts would be effective was also raised, with staff suggesting that eight weeks was too long, as they felt that effectiveness was diminishing “after a few weeks”. It was highlighted that practitioners get many clinical alerts and “the constant having to click the screens off before you can get started” can leave practitioners frustrated and reduce effectiveness.

The practitioners fed back that the training had built their awareness of health literacy; “it really made me think about when I’m in a consultation with patients that actually they might not always understand”. Examples were given of improved practice in communication with patients in the consultation: “it may well have made me think about how … I spoke to the patient … medical jargon”. Perceived gaps in the training were the lack of information about how health literacy differs from learning disability, and the need to include building skills to discuss health literacy with patients. In addition, some practitioners did not understand that the alerts were based on a person’s likelihood, from their socio-demographic characteristics and area of residence, to be below the health literacy threshold, rather than being diagnostic. One participant said “one time the alert came up for a teacher and I thought—that can’t be right…”. Another practitioner, however, recognised that patients with higher everyday life skills might still have low health literacy: “we start spouting off really, you know, medical terminology, different procedures, and things, then actually, unless you work in that area, no matter how intelligent you are, you’re not actually going to know what those terms mean”.

One practice received training via webinar. Staff from this practice stated that, whilst the training had been valuable, they would have preferred face-to-face teaching. 

### 3.4. Practitioner Suggestions for Addressing the Issues Identified

To help resolve the concern that patients might see, and be offended by, the alerts, practitioners suggested that the wording be tailored to focus on the practitioner rather than the patient. Suggestions for the improved wording included “consider practitioner communication skills”. The practitioners felt that improving the wording would lead to the alerts becoming a “positive thing that’s a reminder (to the practitioner)”, as opposed to something which makes the clinicians “uncomfortable or the patient awkward”.

Suggestions for improving training included clarification about the difference between low health literacy and learning disability, and inclusion of role play training on how to have discussions with patients who might notice the pop-ups on their screens. Practitioners also suggested that the training included explanations about how the alerts are generated, i.e., that the patient socio-demographic data and area of residence are used to generate a risk score, which is an indicator rather than being ‘diagnostic’.

Practitioners suggested that one approach to reducing the risk of the alerts upsetting patients was for clinical staff to open the patient screens and acknowledge and close any clinical alerts before the patient was called into consultations. Some practitioners felt that this is would not always be possible, citing time constraints. Another suggestion to reduce the risk of patients feeling offended by the alerts was to take a ‘whole practice’ approach to the intervention. It was suggested that information could be prominently displayed throughout the practice, explaining that the practice is seeking to improve communication with patients, and that this would include practice training and reminders to practitioners about communication during some consultations. 

#### Feedback from Voice Research Support Group

The results of the study were presented and discussed, as was the feedback from the health care practitioners. The patient and carer group fed back that, since the study was focused on improving the consultation experience for patients with low health literacy, patient recruitment would be best conducted with telephone contact rather than by written invitation. The group supported the practitioner views on improving the training, improving the wording of the clinical alerts, and adopting a practice-wide approach as part of the intervention.

## 4. Discussion

### 4.1. Key Results

The rapid review process ensured that both components of the intervention were in line with current evidence and designed to maximise the effectiveness of the intervention. Key factors in maximising the effectiveness of the educational component were to ensure multi-component training involving small group work and role-play, identification of barriers to change and development of plans to overcome those barriers, and involvement of a local opinion-leader in the training. Key factors in maximising the effectiveness of the clinical alerts were the development of stand-alone, automated, ‘push’ alerts that appeared in the consultation and required acknowledgement from the clinician. The alerts only appeared for patients at high risk of low health literacy and only in consultations where additional time was given and the focus of the consultation was on building patient capacity to self-manage long-term health conditions.

Four practices were recruited (33% of eligible practices). Feedback from clinicians was that the intervention had increased their knowledge and skills in relation to health literacy. Feedback included suggestions for improving the intervention, in particular training on how health literacy differs from learning disability, and how to have discussions with patients around health literacy. The feedback from the practitioners also indicated that more information should be given about how the alerts are generated and what they indicate, i.e., that they are not ‘diagnostic’ of low health literacy but rather act as a reminder to the practitioners to use the knowledge and skills acquired through training, and will appear when patients are at greater risk of having low health literacy in view of their socio-demographic characteristics and where they live. As described above, the training had been shortened to 3 h to make it more attractive to busy practitioners, but this feedback indicates that a full day of training may be required to cover all the relevant issues. 

Suggestions for improving the clinical alerts were to improve the wording to focus on practitioner skills rather than patient skills and to reduce the period of time during which the alerts would appear from the current eight weeks, although no specific time was suggested. A key suggestion was that the intervention should involve the whole practice, with information about health literacy and the purpose of the training and clinical alerts being to improve practice staff communication and consultation skills. 

### 4.2. Strengths and Limitations

The project used an evidence-based step-wise approach to developing the intervention, including building on current knowledge through rapid evidence reviews, and post-hoc feedback from clinicians and a patient and carer group. 

A key limitation is that, whilst health literacy is a holistic concept that reflects a dynamic balance between the capacities of individuals and the health and wider societal settings in which they are living, the algorithm used in this project was based on the only health literacy data available in England, i.e., functional health literacy. Functional health literacy is correlated with more holistic health literacy capacities, but whilst the correlation is statistically significant, it is only moderate in size [36]. It should be noted that the algorithm results in a risk score reflecting the capacities measured in the baseline data; where data capturing a wider range of skills are available, the risk scores produced will reflect this.

As this was a development project, no quantitative assessment of the impact of the intervention on clinical communication skills or patient outcomes was undertaken. Assessments of the revised intervention should include more detailed practitioner feedback on the impact of the training on their clinical communication skills, and what ‘added value’ the alert component brings. The current recommendation is for a ‘universal precautions’ approach to health literacy practice, i.e., the provision of high-quality care to all irrespective of health literacy capacities [37]. Whilst practitioners gave some evidence that the alert acted as a reminder to use the skills built through the training, evaluation of the revised intervention should explore this in more depth. Feedback of the impact of the intervention on the patient experience should also be gathered. 

### 4.3. How This Research Links to Current Knowledge

To our knowledge, this is the first intervention to use a combination of clinical training and automated clinical alerts on electronic health records to improve clinical health literacy skills and practice. 

### 4.4. Interpretation

The rapid review process ensured both components of the intervention were evidence based. Post-hoc feedback from clinicians and patients and carers further improved the intervention. 

The one practice that received web-based training stated that they would have preferred face to face training; this may have implications for scaling up any future interventions. 

Key limitations identified in this study should be explored in studies of the revised intervention. 

## 5. Conclusions

Health care practitioners have a key role in developing the health literacy responsiveness of health systems to ensure that they address the health literacy needs of the people they serve. The approach to intervention development taken in this project has resulted in an evidence-based intervention that has been further improved by feedback from clinicians who undertook the training and from a patient and carer group. Further work to develop the intervention will be undertaken through more feasibility testing followed by a pilot- and then full-randomised controlled trial. Since the aim of the intervention is to improve the health literacy awareness, skills and practice of health care practitioners, suitable measures of change will be those that capture patient views on practitioner communication, such as those used in the NHS England GP Patient Survey [38].

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
