# Peer review of "The Evidence-Based Development of an Intervention to Improve Clinical Health Literacy Practice"

_ijerph, 2020, doi:10.3390/ijerph17051513_

Round 1

Reviewer 1 Report

Thank you for your work in this important area. Primary care has a significant role to play in responding to the health literacy needs of the population.

The evidence-based approach to training and engaging with the primary care doctors and nursing staff is laudable and the use of existing technology such as the EHR functionality provides useful insights about the acceptability and long-term sustainability of this kind of "nudging". 

My main concern is the way that health literacy is operationalised within your study. The definition provided suggests a broad view of health literacy that takes into account the context of the everyday life of people. However, the intervention has a much narrower focus, particularly on literacy and numeracy competencies. While health literacy may have a social gradient, when considered in a broader sense, it also cuts across the social gradient. People have various health literacy needs and preferences and these are dynamic. The feedback from a practitioner (line 214-15) that a teacher could not have low health literacy may be indicative of a lack of understanding in this regard. While it is suggested in your paper that this is indicative that the practitioner did not understand that the algorithm was not diagnostic, it also is indicative of a narrow view of health literacy that stems from strongly linking health literacy to literacy and numeracy skills. This is further indicated in relation to the confusion between health literacy and learning disability (line 225). 

The algorithm used to identify those most at risk is focusing primarily on the social gradient. Many people with health literacy needs may not be identified through this process. The diversity of health literacy needs that extend beyond literacy and numeracy is not identified through this process. This needs to be mentioned as a limitation of the study.

It would be interesting to know what percentage of patients were identified by the algorithm and with a long consultation for chronic conditions. If this percentage is high, then perhaps this could be indicative of a need for a universal precautions approach (and indeed previous research indicates that HL should be a consideration in every consultation). And rather than a clinical alert based on the social gradient algorithm, perhaps what is required is an alert for the patients with longer consults for chronic conditions. Further research is required in this area. 

It would be useful to know what kind of activities were undertaken in the training. For example, was teach back used in the role play? This kind of approach emphasizes the role of the practitioner and their skills in responding to the needs of the patient. This is more in line with the suggestions for improving the wording of the clinical alert to be more focused on the practitioner response than the patient's "problem". And as, your study has found, to be potentially less stigmatizing. 

It is not clear from your paper how the practitioners responded to the alert - how did they change their practice? Or was it more only awareness. While the paper states (line 105-7) that the training was based on content already shown to improve confidence and knowledge, it is not clear whether in your study this translated to improved practice and skills - I note that this is an area for further research. However, you have reported on some impact of the training (line 202-215), was there any mention of practice change in this feedback? 

To address these concerns, the introduction needs to clearly explain the difference between the algorithm use to identify those at risk of low health literacy and health literacy. Further that the algorithm is not diagnostic, therefore it does not provide practitioners with clues as to how or in what ways to respond to health literacy needs specific to the patient, but rather, prompts their health literacy awareness more generally. 

Further testing and improvements to the intervention could include the incorporation of concepts such as health literacy responsiveness, and ways to better understand and respond to the diversity of health literacy needs and preferences. 

Author Response

My main concern is the way that health literacy is operationalised within your study. The definition provided suggests a broad view of health literacy that takes into account the context of the everyday life of people. However, the intervention has a much narrower focus, particularly on literacy and numeracy competencies. While health literacy may have a social gradient, when considered in a broader sense, it also cuts across the social gradient. People have various health literacy needs and preferences and these are dynamic. The feedback from a practitioner (line 214-15) that a teacher could not have low health literacy may be indicative of a lack of understanding in this regard. While it is suggested in your paper that this is indicative that the practitioner did not understand that the algorithm was not diagnostic, it also is indicative of a narrow view of health literacy that stems from strongly linking health literacy to literacy and numeracy skills. This is further indicated in relation to the confusion between health literacy and learning disability (line 225). The algorithm used to identify those most at risk is focusing primarily on the social gradient. Many people with health literacy needs may not be identified through this process. The diversity of health literacy needs that extend beyond literacy and numeracy is not identified through this process. This needs to be mentioned as a limitation of the study.

Response: Thank you for this insightful and important comment. The background has been expanded to explain the difference between the holistic definition given in the paper and the fact that the algorithm is based on functional skills, and the discussion widened to include the implications of this. 

It would be interesting to know what percentage of patients were identified by the algorithm and with a long consultation for chronic conditions. If this percentage is high, then perhaps this could be indicative of a need for a universal precautions approach (and indeed previous research indicates that HL should be a consideration in every consultation). And rather than a clinical alert based on the social gradient algorithm, perhaps what is required is an alert for the patients with longer consults for chronic conditions. Further research is required in this area. 

Response: Thank you. We did not measure the percentage of the patients who were identified by the algorithm, but we can be confident it would have been more than the national UK average of 61%. Indeed, this was one of the main reasons for restricting the alerts to patients attending for in-depth chronic health conditions review, where health literacy capacities are a key focus, rather than all consultations. The point about a universal precautions approach is well made, and any future study should include practitioner feedback on whether, and if so how, the alerts add anything over and above the training. We have added these points to the discussion.

It would be useful to know what kind of activities were undertaken in the training. For example, was teach back used in the role play? This kind of approach emphasizes the role of the practitioner and their skills in responding to the needs of the patient. This is more in line with the suggestions for improving the wording of the clinical alert to be more focused on the practitioner response than the patient's "problem". And as, your study has found, to be potentially less stigmatizing.

Thank you. The training included teach-back. This has been added to the text.

It is not clear from your paper how the practitioners responded to the alert - how did they change their practice? Or was it more only awareness. While the paper states (line 105-7) that the training was based on content already shown to improve confidence and knowledge, it is not clear whether in your study this translated to improved practice and skills - I note that this is an area for further research. However, you have reported on some impact of the training (line 202-215), was there any mention of practice change in this feedback? 

Response: Thank you for raising this important point. There were data feeding back on changes in clinical practice and these are now incorporated into the text.

To address these concerns, the introduction needs to clearly explain the difference between the algorithm use to identify those at risk of low health literacy and health literacy. Further that the algorithm is not diagnostic, therefore it does not provide practitioners with clues as to how or in what ways to respond to health literacy needs specific to the patient, but rather, prompts their health literacy awareness more generally. Thank you. We believe we have addressed this through our changes detailed above.

Further testing and improvements to the intervention could include the incorporation of concepts such as health literacy responsiveness, and ways to better understand and respond to the diversity of health literacy needs and preferences.

Response: This is an important point but does not arise from our data. We have added it into the conclusion.

Reviewer 2 Report

This well-written and fluent manuscript describes the feasibility study of a two-components intervention, which is addressed to increase health literacy awareness and skills in general practices. Two components are GP and nurses training in health literacy, followed by the implementation of a system of pop-up alerts appearing on the GP clinical practice screen.

In this study, there is no control group and there is not any quantitative measurement of the intervention’s effectiveness. However, the authors are planning a second stage to perform a pilot- and RCT study based on the results of this preliminary feasibility study.

I have some minor concern:

Line 81: explain what it is meant for “rapid” review

Line 89: please, include the references for two key papers

Line 132: the acronym HL should be explained

Line 201: correct the typo (Issued)

Line 287: I think that the authors should sketch the main characteristics of the RCT they plan to do (arms, end-points and so on).

Appendix 1-2: they are difficult to understand. Please, include captions and line numbers.

Author Response

I have some minor concern:

Line 81: explain what it is meant for “rapid” review.

Response: This has been added, with a supporting reference.

2. Line 89: please, include the references for two key papers.

Response: The reference for the rapid review methodology has been added, and the references for the key papers (one per rapid review) are already included in the ext. We hope this answers your point.

3. Line 132: the acronym HL should be explained.

Response: Apologies the acronym “HL” has been removed throughout.

4. Line 201: correct the typo (Issued)

Response: Thank you this has been done.

5. Line 287: I think that the authors should sketch the main characteristics of the RCT they plan to do (arms, end-points and so on).

Response: Thank you for this suggestion; there is, however, considerable further development to be undertaken on the intervention before a full RCT is designed, so we feel it is too early to map out the characteristics of a future RCT. We have, however, clarified the aim of the project (to improve practitioner skills and hence patient rating of GP communication skills) and have suggested that a validated questionnaire capturing that would be suitable as a main outcome measure in a future RCT. We hope this is satisfactory. 

6. Appendix 1-2: they are difficult to understand. Please, include captions and line numbers.

Response: We have added explanatory titles to the columns in these appendices and hope that these are now clearer.